# Effects of Flood on Phytoplankton Diversity and Community Structure in Floodplain Lakes Connected to the Yangtze River

**Yutao Wang [1], Zhongya Fan [2], Wencai Wang [2], Zhongze Zhou [1] and Xiaoxin Ye [1,\*]**

1   School of Resources and Environmental Engineering, Anhui University, Hefei 230601, China; wangyutao323@163.com (Y.W.); zhzz@ahu.edu.cn (Z.Z.)

2   State Environmental Protection Key Laboratory of Water Environmental Simulation and Pollution Control, South China Institute of Environmental Sciences, Ministry of Ecology and Environmental of PRC, Guangzhou 510530, China; fanzhongya@163.com (Z.F.); wangwencai@scies.org (W.W.)

\*   Correspondence: yexx@ahu.edu.cn

**Abstract:** Flood events can significantly affect the physical and biological processes of aquatic ecosystems in a short time, leading to rapid changes in phytoplankton community structure. The Huayanghe Lakes experienced extreme flooding in the summer of 2020, with the water level reaching 16.42 m. In order to understand the effects of flooding on phytoplankton diversity and community structure, eight samples were collected in the Huayanghe Lakes from 2019 to 2020. Water-level disturbance has a significant influence on lake-water quality and phytoplankton community structure. The results showed that the Secchi depth increased from 65.36 to 8.52 cm, while the concentration of total nitrogen (from 0.98 to 0.7 mg/L) and total phosphorus (from 0.04 to 0.031 g/L) decreased. In addition, flooding significantly increased the Shannon–Wiener diversity index, the Pielou index, and the Margalef richness index by an average of 43.5%, 36.7%, and 40.21%, respectively. The phytoplankton community structure in the Huayanghe Lakes changed due to the change of physicochemical environment caused by flood. While in the pre-flood period phytoplankton was composed of large diatoms (e.g., *Aulacoseira granulata*), cyanobacteria (e.g., *Microcystis* sp., *Anabaena* sp., and *Aphanizomenon* sp.) and other multicellular taxa, the flood period showed an increase in the proportion of chlorophytes and diatoms that quickly adapted to settle in new environments. Pearson correlations and redundancy analyses showed that water level fluctuation was the most significant environmental factor affecting the phytoplankton community between the regular hydrological cycle and flood periods. There are few studies on phytoplankton in the Huayanghe Lakes, and the present study provides basic data on phytoplankton diversity and community structure. In addition, it provides a theoretical basis for controlling water level change in the Yangtze River.

**Keywords:** flood; phytoplankton diversity; community structure; Yangtze River

## 1. Introduction

In recent years, under the influence of global climate change, extreme hydrological events such as floods have occurred more frequently, and the effect of floods on aquatic ecosystems has also become important. Floods are extreme hydrological events that can affect the structure, function, and dynamics of lake ecosystems [1,2]. Floods cause different changes in the lake water level, resulting in changes in the species composition, abundance, and biomass of phytoplankton in the lakes. The interaction between phytoplankton and zooplankton is greatly influenced by the physicochemical factors of lakes caused by flood or drought [3]. The Huayanghe Lakes we studied are typical shallow lakes connected to the Yangtze River. Surprisingly, although little research has been done on the phytoplankton community in Huayang Lakes, it plays an important ecological role in the Yangtze River Basin. This knowledge is essential for understanding and predicting the impact of water-level fluctuations on lake aquatic communities. The region experienced rainy seasons in the summer of 2020, and the water level has now reached 16.42 m.

The phytoplankton community structure is mainly affected by environmental factors such as physical, chemical, and biological characteristics, and hydrodynamics [4]. Among them, nutrients and light are the main environmental factors affecting phytoplankton growth [5]. Hydrodynamics, hydrology, and other factors affect phytoplankton diversity and nutrients, which directly or indirectly affect the phytoplankton community structure [6]. Flooding has direct and indirect effects on phytoplankton: on the one hand, phytoplankton is lost directly due to flooding, and their biomass, abundance, and species diversity are affected. On the other hand, flood changes a lake's physical and chemical factors indirectly affect the phytoplankton community structure.

In order to study the effects of extreme flooding on the phytoplankton community in the Huayanghe Lakes, we evaluated the phytoplankton diversity and community structure. We hypothesized that lake nutrient concentrations and phytoplankton biomass would decrease during the flood periods. Furthermore, we hypothesized that the phytoplankton community structure differs, and dominant species may change between the regular hydrological cycle and the flood period; the given results will be compared with phytoplankton results from other years.

## 2. Materials and Methods

### 2.1. Study Area

The Huayanghe Lakes (116°00″ E–116°33″ E, 29°52″ N–30°58″ N), which consist of four lakes, namely, Lake Longgan, Lake Daguan, Lake Huang, and Lake Po, are located in the north of the middle reaches of the Yangtze River in Anhui Province, China (Huangda Lake is composed of Lake Daguan and Lake Huang). The lake district has a subtropical climate with four distinct seasons, rain fall mostly occurs in the summer [7]. The total area of the Huayanghe Lakes is about 966 km$^2$ when the maximum depth of lakes is 17 m. Sampling points are shown in Figure 1.

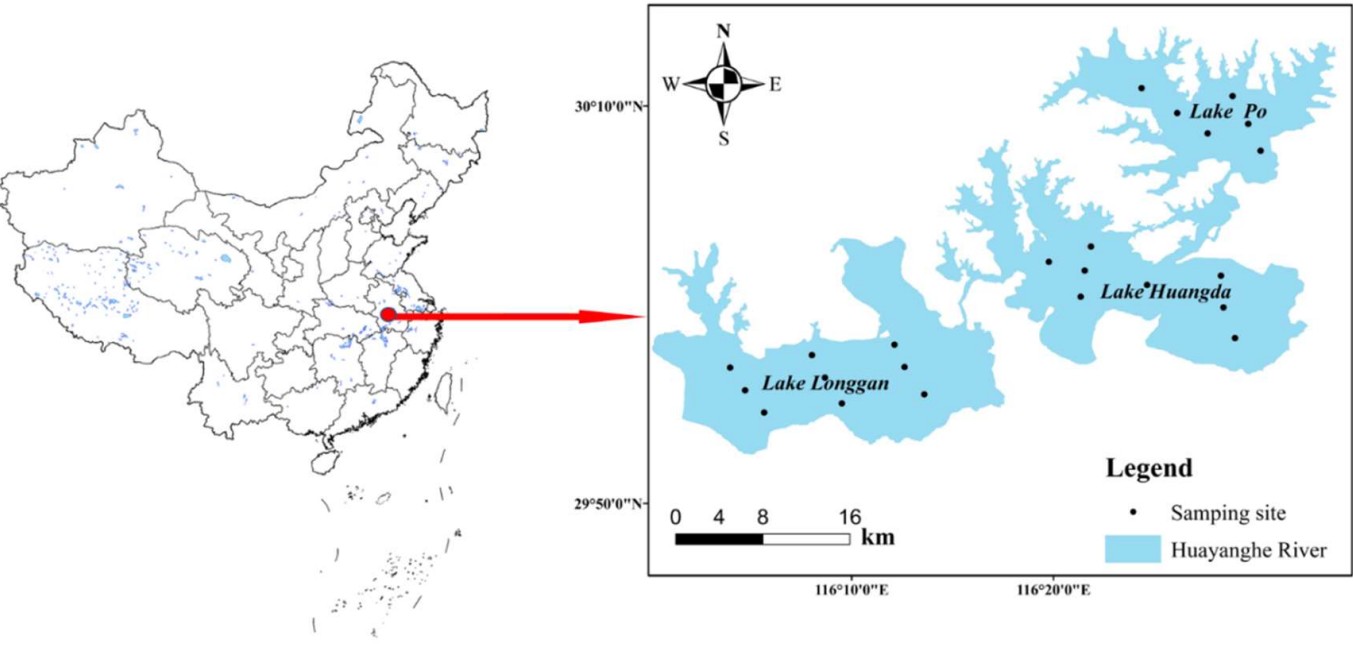

**Figure 1.** Map of the study area and sampling sites.

We sampled 23 sites in four lakes during the period of January 2020 to October 2021. The area experienced extreme rainfall in the summer of June 2020. The water level of HYH reached a new maximum of 16.42 m (Figure 2, data from www.ncc-cma.net. Date of visit is 2 December 2021). The sampling sites are shown in Figure 1.

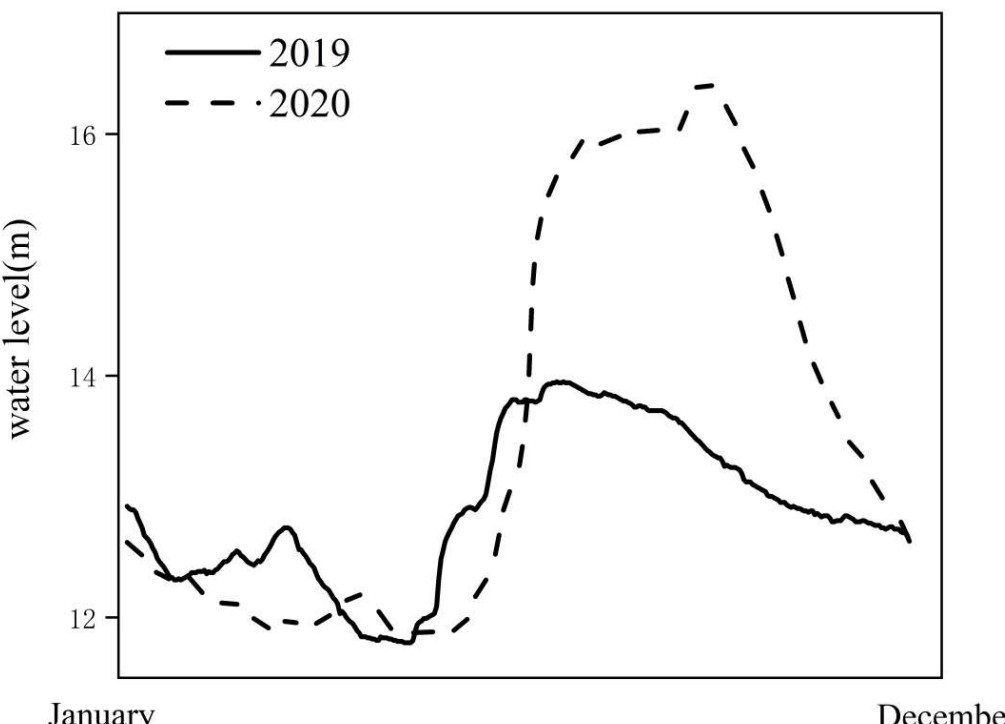

**Figure 2.** Seasonal variation of water level in Huayanghe River. The solid and dashed lines denote the daily average rainfall in 2019 and 2020, respectively.

### 2.2. Sampling

We measured the water temperature, pH, conductivity, and dissolved oxygen using the Hach HQ40d portable multimeter in the field. We measured the Secchi depth with the Secchi disk. Total nitrogen (TN) and total phosphorus (TP) were analyzed according to standard methods (SEPB 2002) [8].

In the lab, phytoplankton samples were stored in 1-litre plastic bottles and then mixed with 1% Lugol iodine solution and left for at least 48 h. After draining the supernatant with a 2 mm diameter hose, the preserved sample was concentrated to 30 mL [9,10]. Then, 0.1 mL samples were counted in a counting chamber under a light microscope (Olympus, BX53) [11]. Each specimen was identified and counted. The biovolume (mm$^3$/L) of every phytoplankton specie was calculated based on the solid geometric shape and the conversion of biovolume into biomass was done. The identification of the phytoplankton species was carried out according to the method used by Hu and Wei [12]. The species were grouped into functional groups as described by Reynolds [13].

### 2.3. Data Analyses

In accordance with the Shannon–Wiener index (H′) [14], the Pielou index (J′) [15], and the Margalef index (d) [16], we investigated the species diversity as follows:

$$H' = -\sum_{i=1}^{s} \frac{n_i}{N} log_2 \frac{n_i}{N}$$

$$J' = H'/lnS$$

$$d = (S-1)/lnN$$

where $n_i$ is the number of individuals of species $i$, N is the total number of individuals of all species, S is the total number of phytoplankton species, and $n_i/N$ represents the relative proportion of species $i$.

A statistical analysis was carried out using IBM SPSS Statistics (Version 20.0). Before the analysis was carried out, the data were log transformed to ensure the data were normally distributed. Variations of physical and chemical indexes and the phytoplankton biomass in different periods were analyzed using one-way ANOVA. Redundancy analysis (RDA) was used to evaluate the effects of environmental variables on the phytoplankton community. The maximum gradient length of the DCA ordination axis is less than 3; thus, RDA was appropriate for the analysis. RDA analysis was performed using Canoco 5 software [17]. We used Origin for the columnar graphs. The region map was drawn by inverse distance weighting method [18,19] and ArcGIS software (Version 10.4, ESRI).

## 3. Results

### 3.1. Environmental Characteristics Parameters

The samples were collected from 26 sampling sites from January 2019 to October 2020. From June 2020, the water depth of the lake increased with the increase of rainfall, exceeding 11.88 m. Then, the flooding began. The water-level in HYH is between 9 and 17 m, with similar seasonal characteristics, the highest in the summer and the lowest in the winter (Figure 2). Floods occurred in the summer of 2020, and the water depth increased by 1.85 m compared with other years. SD and WD increased significantly during the flood period (by 22.8% and 38.9%, respectively). SD and WT, as well as WD, had an increasing and then a decreasing trend. The total nitrogen concentration decreased from 0.98 to 0.7 mg/L, and the total phosphorus concentration decreased from 0.04 to 0.031 g/L. The concentrations of TN, TP, and turbidity were notably ($p < 0.05$) lower in the flood period than in the regular hydrological cycle (Figure 3). Flooding occurs mainly in the summer, so this part focuses on comparing the differences between the summers in different years.

### 3.2. Phytoplankton Composition and Biomass

A total of 125 species of phytoplankton were observed in this study. Of these, 76 species of phytoplankton were in the regular hydrological cycle, and 89 species were found in the extreme hydrological cycle. In the regular hydrological cycle, cyanobacteria and bacillariophyta always dominated the phytoplankton community (Figure 4). As the water level rose, the proportion of bacillariophyta and chlorophyta increased. The biomass of phytoplankton during the regular hydrological cycle was 33.34 ± 3.29 mg/L and 11.96 ± 0.26 mg/L during the flood period. The changes of physical and chemical environment caused by flooding changed the phytoplankton community structure. *Microcystis* sp., *Anabaena* sp., and *Aphanizomenon* sp. were the dominant species in the flood period. They were both multicellular populations.

The phytoplankton diversity indices (H′, J′, and d) of the conventional hydrological cycle and the extreme hydrological cycles were significantly different ($p < 0.05$, in most cases). Flooding increased H′, J′, and d by an average of 43.5%, 36.7%, and 40.21%, respectively (Figure 5).

### 3.3. Correlations between Phytoplankton Community Composition and Environmental Variables

RDA was used to evaluate the relationship between phytoplankton dominant species and environmental factors. The RDA ranking diagram includes dominant phytoplankton species and environmental variables. The eigenvalues for RDA axis 1 (0.573) and axis 2 (0.069) explained 59.25% of the variance in the phytoplankton functional groups during the extreme hydrological cycle (Table 1). However, the eigenvalues of the first two axes (0.433 and 0.034, respectively) of the RDA only explained 46.73% (Table 2) of the variance in the phytoplankton taxonomic communities during the regular hydrological cycle (Figure 6). We found that pH, WL, SD, Cond, DO, Turb, TN, and TP were the main factors that affected the phytoplankton taxonomic composition.

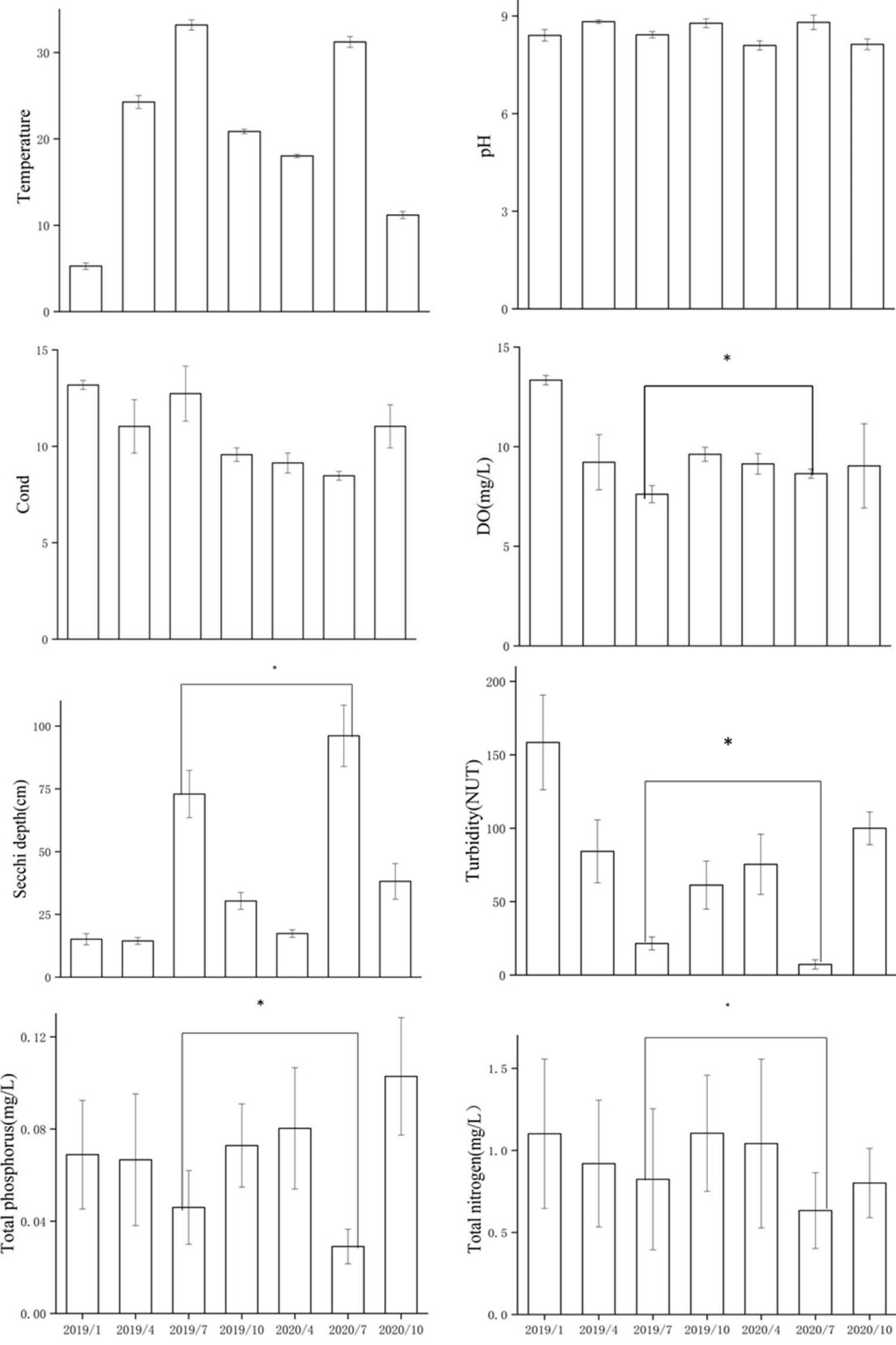

**Figure 3.** The characteristics of environmental factors before and after the flooding of Huayanghe River. Abbreviations used in the diagram: Cond: conductivity, DO: dissolved oxygen (flooding occurred in July 2020) (* $p < 0.05$).

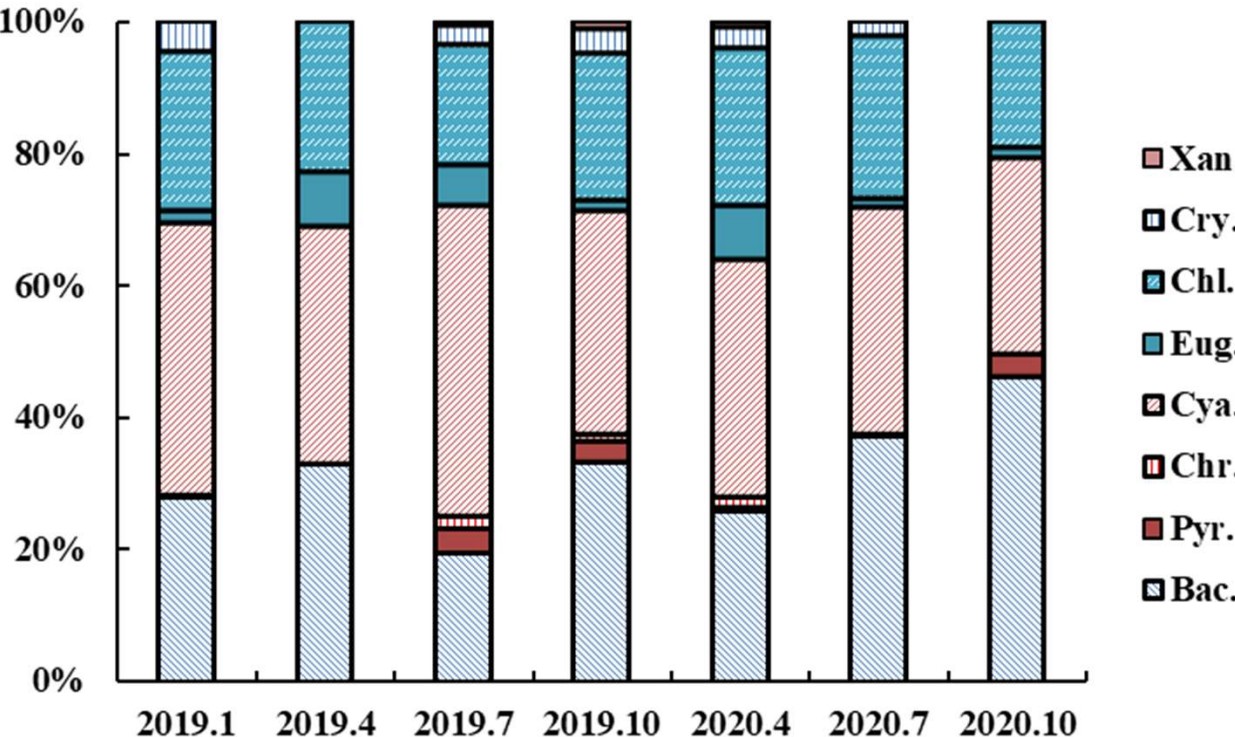

**Figure 4.** Changes in relative densities of different phytoplankton taxa during sampling periods. Abbreviations was used in the diagram: Bac: bacillariophyta, Cry: cryptophyta, Pyr: pyrrophyta, Cya: cyanobacteria, Chr: chrysophyta, Chl: chlorophyta, Eug: euglenophyta, and Xan: xanthophyta.

**Table 1.** Summary of RDA analyses with environment variables explaining phytoplankton assemblage before the flooding of Huayanghe Lakes.

| Axes | 1 | 2 | 3 | 4 | Total Variance |
|---|---|---|---|---|---|
| Eigenvalues | 0.433 | 0.034 | 0.0737 | 0.0479 | 1 |
| Species–environment correlations | 0.9777 | 0.8479 | 0.7175 | 0.7201 | |
| Cumulative percentage variance of species data | 34.89 | 46.35 | | | |
| Cumulative percentage variance of species–environment relationship | 54.16 | 71.95 | | | |
| Sum of all eigenvalues | | | | | 1 |

**Table 2.** Summary of RDA analyses with environment variables explaining phytoplankton assemblage after the flooding of Huayanghe Lakes.

| Axes | 1 | 2 | 3 | 4 | Total Variance |
|---|---|---|---|---|---|
| Eigenvalues | 0.573 | 0.069 | 0.048 | 0.03 | 1 |
| Species–environment correlations | 0.812 | 0.747 | 0.615 | 0.708 | |
| Cumulative percentage variance of species data | 53.72 | 58.51 | | | |
| Cumulative percentage variance of species–environment relationship | 83.39 | 90.82 | | | |
| Sum of all eigenvalues | | | | | 1 |

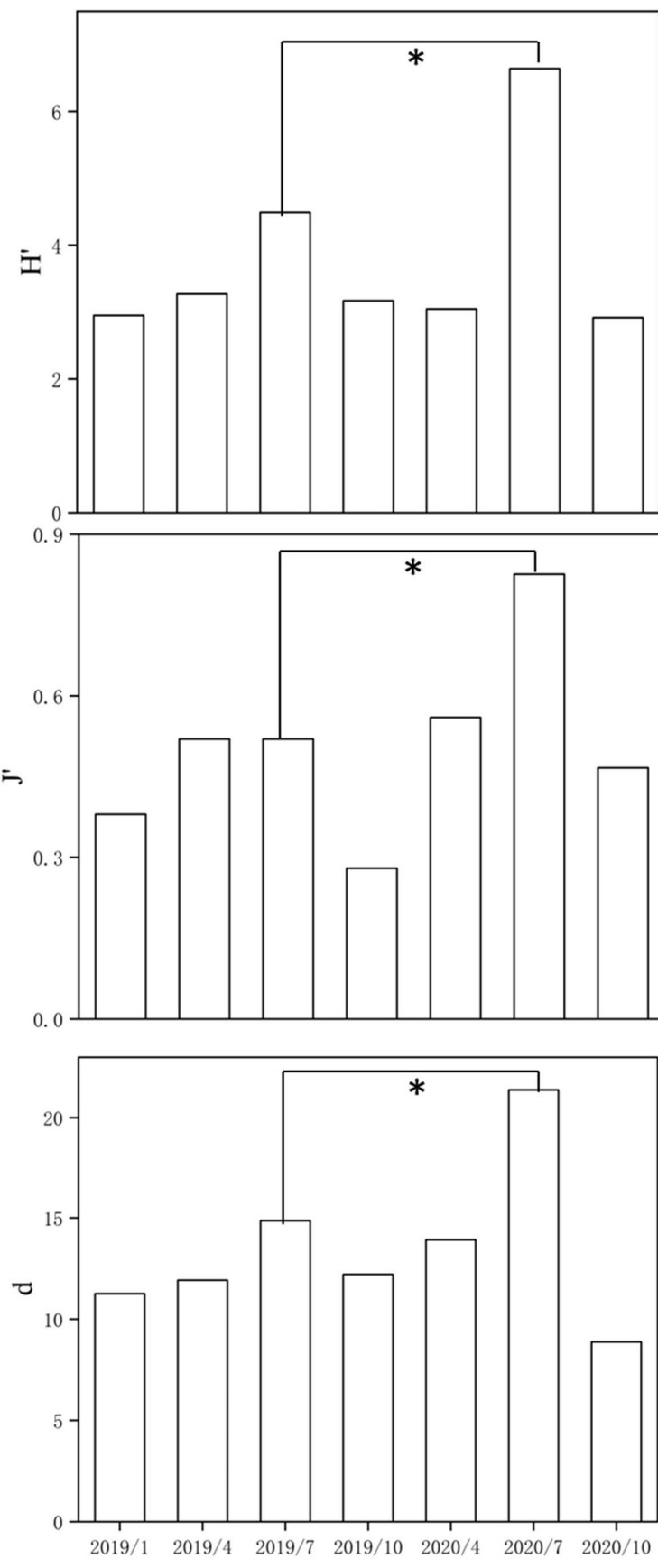

**Figure 5.** Shannon–Wiener diversity (H′), Pielou evenness (J′), and Marglef richness (d) in the Huayanghe Lakes (* $p < 0.05$).

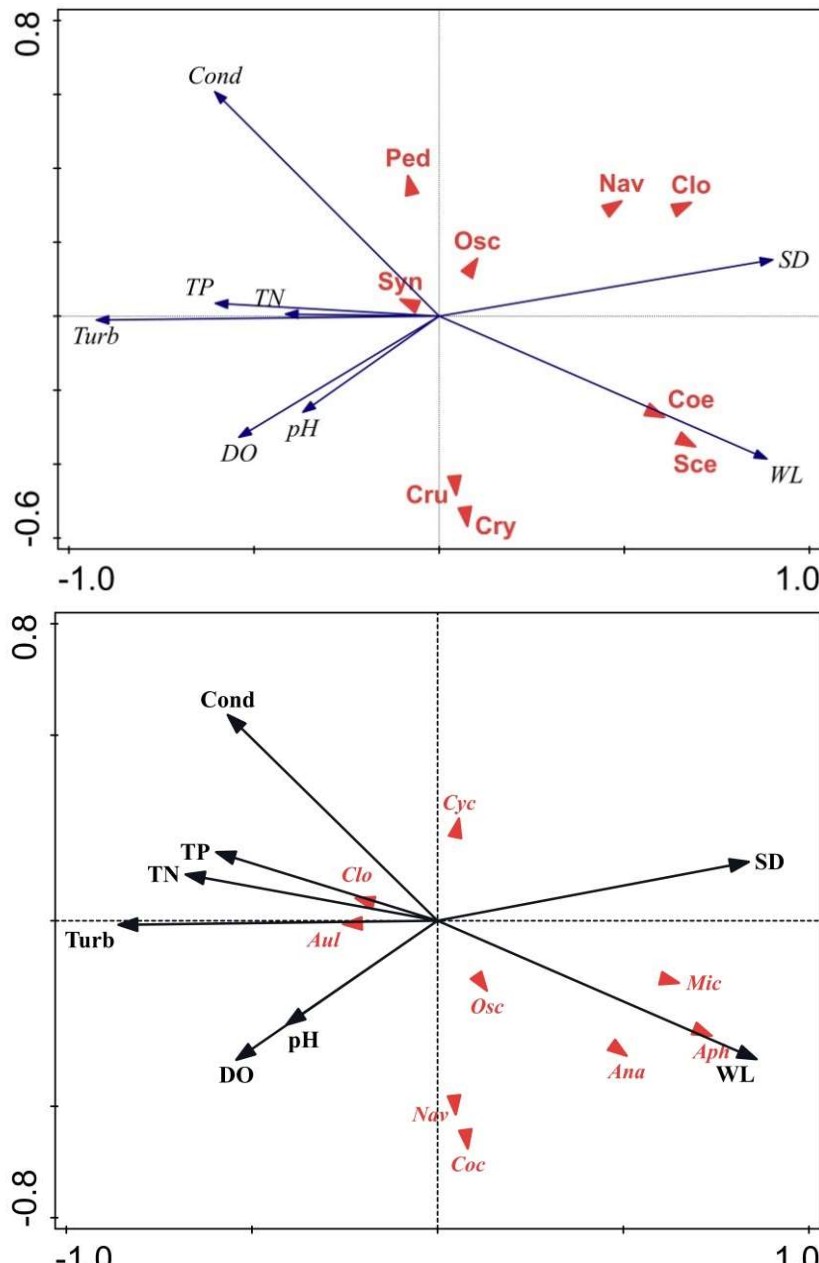

**Figure 6.** Phytoplankton species (triangle symbol) and environmental factor (black lines with arrowhead) biplot based on RDA before and after the flooding of Huayanghe Lakes. Abbreviations used in the figure: WL, water level; SD, Secchi depth; Cond, conductivity; DO, dissolved oxygen; TN, total nitrogen; TP, total phosphorus; and Turb, turbidity. Taxonomic groups: Cry, *Cryptomonas ovata*; Cru, *Crucigenia* sp.; Coe, *Coelastrum reticulatum*; Sce, *Scendesmus* sp.; Clo, *Closterium* sp.; Nav, Navicula sp.; Osc, *Oscillatoria* sp.; Syn, Synedra sp.; Ped, *Pediastrum* sp.; Cyc, *Cyclotella* sp.; Aul, *Aulacoseira granulata*, Ooc, *Oocystis lacustris*; Ana, *Anabaena* sp.; Mic, *Microcystis* sp.; and Aph, *Aphanizomenon flosaquae* (the regular hydrological cycle is shown above, and the flood period is shown below). During the regular hydrological cycle, the relative density of *Navicula* sp., *Closterium* sp., *Coelastrum reticulatum*, and *Scendesmus* sp. were positively associated with SD and WL. Other dominant species—*Crucigenia* sp., *Cryptomonas ovata,* and *Cryptomonas erosa*—presented a notably positive relationship with Cond. *Closterium* sp., *Aulacoseira granulata*, *Navicula* sp., *Cocystis lacustris,* and *Cyclotella meneghiniana* were related to Turb, TN, and TP. *Navicula* sp. and *Oocystis lacustris* were related positively to DO and pH, the same as *Microcystis* sp., *Aphanizomenon* sp., and *Anabaenopsis* sp. in the extreme hydrological cycle.

## 4. Discussion

### 4.1. Floods Drive Changes in Huayanghe Lakes

Water level fluctuations caused by flooding greatly affect aquatic ecosystems. The physical and chemical parameters of lakes are significantly affected. Therefore, water level fluctuation in floodplain lakes has significant implications [20]. Flooding brings sediment particles from surrounding areas into the lakes, changes the state of sediment, and affects the nutrient cycling process of lakes. Flooding reduced the internal nutrient cycle and led to the exogenous nutrient input becoming the main nutrient source of the lakes [14]. Consequently, there were changes in the environmental physical and chemical factors of the lakes.

Our results showed that the water quality of the flood period was better than in the regular hydrological cycle, which was consistent with the results of Zhang et al. [21] in the Pearl River Delta. This improvement may be related to the dilution effect of increased rainfall. B-Beres et al. [22] found that continuous rainfall led to an increase in river flow and a decrease in nutrient concentrations in lakes. In the flood period, the water level rose rapidly, and the turbidity, total nitrogen, and total phosphorus concentration of lakes decreased. This may have been due to the dilution effect of high water levels. Water level fluctuation is considered to be an important factor in lake nutrient status [23].

### 4.2. The Influence of Flood Events on Phytoplankton Diversity and Community Structure

Changes in phytoplankton community structure are mainly determined by available resources in lakes. Dissolved nutrients and light conditions in lakes are important factors affecting phytoplankton community composition [24]. Changes in the water level caused by flooding have a direct impact on phytoplankton community structure through dilution. Compared with the regular hydrological cycle, the proportion of Cyanobacteria in the flood period decreased, and the proportion of pyrrophyta, chlorophyta, and xanthophyta increased. Multicellular populations of phytoplankton possess morphological and physiological characteristics that can survive in stratified water [25]. For example, *Microcystis* sp., *Anabaena* sp., and *Aphanizomenon* sp. were the dominant species in the flood period. Their individual size protects them from predation by most zooplankton and fish, and the buoyancy regulation allows for a quick adjustment of the vertical position in the lake [26–28]. The biomass of diatoms keeps increasing at the rising stage of the water level; they are more suitable for suspension in strong mixed currents due to their structural characteristics. Lan et al. [29] found that *A. granulata* has good adaptability. It can still survive during floods.

The Huayanghe Lakes have a subtropical climate. Water temperature and water depth both showed a trend of increasing first and then decreasing in one year, but there were no significant differences between the regular hydrological cycle and the flood period. In this study, WT and WD were the main factors affecting the phytoplankton community structure. This indicates that there are obvious seasonal changes in the phytoplankton community structure. This is consistent with the conclusions of Nafi'u and Ibrahim [30]. Both phytoplankton diversity and community structure are affected by environmental factors, especially extreme water level changes. In the regular hydrological cycle, the DO and TN levels were the key factors affecting phytoplankton community structure. However, WD was an important factor affecting phytoplankton diversity and community structure in the flood period.

Flooding may have affected the abundance and diversity of phytoplankton [31]. Our study showed that flooding increased the phytoplankton diversity index, which is consistent with the results of Li et al. [32]. Phytoplankton density in the flood period was lower than the regular hydrological cycle season, which is inconsistent with reports from in the East China Sea [33]. Due to the dilution effect, phytoplankton density in the Huayang River decreased during the flood period. In addition, increased lake water also expanded phytoplankton habitats, and flooding introduced new species. This water may have come from nearby farmlands or mountain springs. The strong increase in the water level observed caused a radical shift in the phytoplankton community. While in the pre-flood period

phytoplankton was composed of large diatoms, cyanobacteria, and other multicellular phytoplankton, the flood period showed an increase in the proportion of chlorophytes and diatoms that quickly adapted to settle in new environments.

**Author Contributions:** Writing—original draft, Y.W.; investigation, Z.F.; formal analysi, W.W.; writing—review and editing, Z.Z., X.Y. All authors have read and agreed to the published version of the manuscript.

**Funding:** This research was supported by Joint Research Project for the Yangtze River Conservation (Phase I), China (No. 2019-LHYJ-01-0212, 2019-LHYJ-01-0212-17).

**Institutional Review Board Statement:** Not applicable.

**Data Availability Statement:** The datasets generated during and/or analyzed during the current study are available from the corresponding author on reasonable request.

**Acknowledgments:** We would like to thank Marci Baun from University of California, Los Angeles for editing the paper.

**Conflicts of Interest:** We declare that we have no known competing financial interests or personal relationships that could have influenced the work reported in this paper.

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
