# Peer review of "Effects of Flood on Phytoplankton Diversity and Community Structure in Floodplain Lakes Connected to the Yangtze River"

_diversity, doi:10.3390/d14070581_

Round 1
Reviewer 1 Report
Review of manuscript diversity-1649679 by Wang et al.
The manuscript by Wang et al. explores changes in phytoplankton community structure of Huayanghe Lakes connected to the Yangtze River (China) in relation to seasonal variation of hydrological regime and water chemical parameters. The questions addressed are relevant to phytoplankton ecology in river-lake systems. The data collection is sufficient in temporal scale comprising two consecutive years with relatively stable low-water hydrological period and high-water period caused by seasonal flooding. However, I have some concerns regarding application of multivariate analysis (RDA) and author’s interpretation of RDA results. Also, there are some conclusions, that were not demonstrated by the results, and thereby sound as unjustified. Below, I offer my suggestions, especially for the method section, that authors may wish to consider during revision of their manuscript.
Abstract
Line 11: “rapid succession of phytoplankton community structure” – Awkward phrase. Please, consider revising. For example, “rapid changes in phytoplankton community structure”.
Line 22: Aulacoseira granulata not “granulate”.
Line 23: Please, replace “other multicellular phytoplankton” by “other multicellular taxa”.
Line 24-25: “the flooding period showed an increase in smaller and more competitive chlorophytes and smaller diatoms…” – This is not supported by the results.
Line 26-27: This conclusion is also not supported by the results. According to RDA, factors that showed significant influence on community structure are the same (see fig. 5).
Line 28-29: The last sentence suggests that the study presented has no more value than regional importance.
Introduction
In general way, the introduction is very short, missing of relevant literature on the topic of study and talks about the important factors driving phytoplankton structure very superficially.
Line 34: The first paragraph starts from regional descriptions of the study lakes. What is lacking to understand phytoplankton dynamics in water bodies of subtropical climate? How is important your study in this respect?
Line 49-51: There are many studies on factors driving phytoplankton community structure including flood events. You are talking about two aspects of the influence of floods. What is the difference between direct mechanisms leading to phytoplankton losses in the first aspect and physical factors influencing community structure in the second aspect?
Line 55-58: The hypotheses presented are predictions. You included expectation that phytoplankton diversity is high in the flood period. Why? No theoretical foundation is presented. Moreover, the statement in the introduction (line 49) suggests the opposite. Thus, you need to include into the introduction a clear message of how phytoplankton diversity is influenced by floods. The prediction that “phytoplankton community structure differed between the regular hydrological cycle and the flood period” is a very general. Which differences are you expecting exactly in algal groups or/and total phytoplankton biomass etc.? In the introduction include references that can be a basis to those expectations.
Materials and methods
Line 62: Where is lake Daguan is located on map (fig. 1)?
Line 66: Water level or maximum depth?
Line 71: Why are only 23 sites from 26 presented in fig.1?
Line 73: Please, replace “high” by “maximum”.
Line 83: “We field measured” – grammar problem. Please, consider revising.
Line 85-86: What is the difference between Secchi depth and water depth if they are both were measured by Secchi disk? How water depth was measured really?
Line 90-95: More details are needed how biomass measurements of the total phytoplankton biomass were made. What about taxonomic literature?
Line 107-108: What exactly ANOVA was used for? What does that mean "significant variance"?
Line 109-110: Where is the result of correlation analysis?
Line 112-113: The purpose of log-transformation is quite different from that you mentioned.
Line 114: Reference?
Line 117-118: What this interpolation was used for?
Results
Line 128: Why you did not present here values of TP and TN (only in the abstract)?
Line 129: Where is graph for TN? In fig. 3, there are two graphs for TP.
Line 130: If so, where is ANOVA comparison between summers of different years?
Fig. 3. The graphs are not informative in relation to before- and after-flood periods. Make it clear which date is impacted by flood.
Line 140: “boost the advantages of bacillariophyta and chlorophyta” – Do you mean biomass or density? That is mentioned only once in the results and was not approved by ANOVA, thus cannot be considered as justified result.
Line 142: “Phytoplankton biomass decreased significantly” – This was also not approved by ANOVA results and no graph was presented.
Line 145: Here you say that cyanobacteria were dominant species in flood period. But above bacillariophyta and chlorophyta were “boosted” (line 140). So, what do you mean?
Fig. 5. Please, indicate which RDA-plot is related to before- and after-flooding period.
Line 171: Oocystis lacustris not “Cocystis”.
Line 179: Cryptomonas erosa is not presented on the RDA-plot.
Line 180-181: All these three species (Navicula sp., Oocystis lacustris, Cyclotella meneghiniana) were not related to Turb, TN and TP, because they are located near the center of the 1st axis, that is correlated with these parameters. But they are related to the 2nd axis. Moreover, Navicula sp. and Oocystis lacustris are likely related positively to DO and pH, because direction of arrows for species and factors are coincided on the RDA-plot. The same we can say about Microcystis sp., Aphanizomenon sp., and Anabaenopsis sp. Thus, conclusion about negative relationships between these species and DO and pH on line 182-183 is wrong.
Discussion
Overall, the discussion is written in choppy manner, inconsistent and weakly justified.
Line 187: “physical and chemical parameters are affected” by what?
The main problem of the first paragraph, especially the last two sentences, is linearity in presentation of information.
Line 200: I do not see a considerable improvement of water quality during the flood period. There can be statistically significant (that is not approved) difference. But decrease in TP is rather small and TP values remains in mesotrophic boundaries.
Line 210-211: “Significant” increase in diatom biomass during “rising water level” and increase in cyanobacteria biomass during “flood period” were not presented graphically and statistically approved.
RDA did not show this also because analysis was performed separately for different hydrological periods. Moreover, here is not clear what is compared. What does period of “rising water level” means?
Therefore, I suggest to perform RDA for combined data from both hydrological periods to assess the effect of rising water level on phytoplankton community. Furthermore, total phytoplankton biomass and biomass of the major algal groups can be compared between hydrological periods using ANOVA.
Line 212: What does “other algae” means?
Line 213-214: The sentence is not clear due to grammar problems.
Line 216-217: Individual size protects them (colonial cyanobacteria) from predation by most … fish?
Line 221: “A. granulata has a good adaptability” for what?
Line 223-224: “no significant differences (WT – water temperature(?) and WD) between the regular hydrological cycle and flood period”. But in the next sentence you say: “WT and WD were the main factor that affecting the phytoplankton community structure”. Also, in line 230-232 you claim that different factors were important in hydrologically stable and flood periods. However, this is not supported by the results of RDA analysis. Actually, they are the same (Fig. 5). I suggest to perform partial RDA (Borcard et al., 1992) to assess difference in relative importance of physical and water chemical parameters between hydrological periods.
Line 238-239: What is the physical mechanism of “disappearance of individual organisms”? Moreover, your results show the opposite tendency of the increased species richness during high water level.
Line 240: “flooding could also introduce new species”- What is the source of these new species for lake-river system?
Line 243-244: “the flooding period showed an increase in smaller and more competitive chlorophytes and smaller diatoms” - This is not supported by the results (see comments to line 140).

Author Response
Dear editor and reviewers,
Thank you for your consideration for publishing our research in diversity and providing insightful and constructive comments on our manuscript. We have carefully addressed all comments throughout the manuscript. We take special care to elaborate in the revised manuscript regarding the (1) improving the clarity and accuracy of the expressions, including but not limited to those you pointed out; (2) enriching the some references by comparing our study with similar studies. We believe that both scientific relevance and clarity of the paper have benefited from these comments.
Our detailed answers to the editor and reviewers' comments are provided below in blue. The changes made in the text are also highlighted in blue.
Thank you for your time.
Sincerely yours,
Yutao Wang, Zhongya Fan, Wencai Wang, Zhongze Zhou, XiaoxinYe

Reviewer 2 Report
Thank you for suggesting myself as a reviewer. The authors has attempted a good work and succeeded in finding the results. However, there are some lacunae, which I have shown as comments. I request authors to incorporate the comments in the manuscript and improve the same.

Author Response

(The authors gave the same response as above.)
